# LEARNING PARAMETER SHARING WITH TENSOR DECOMPOSITIONS AND SPARSITY

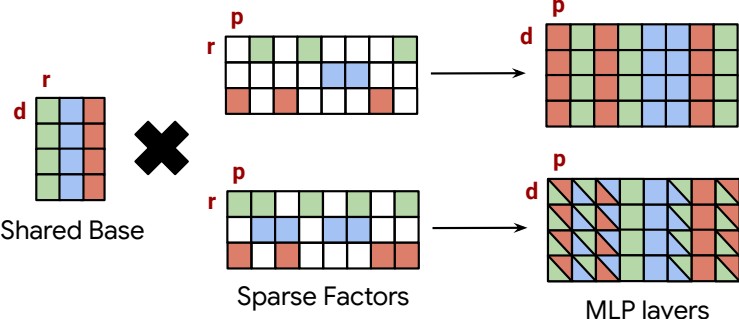

Figure 1: **Fi**ne-grained **P**arameter **S**haring (FiPS).

## ABSTRACT

Large neural networks achieve remarkable performance, but their size hinders deployment on resource-constrained devices. While various compression techniques exist, parameter sharing remains relatively unexplored. This paper introduces **Fi**ne-grained **P**arameter **S**haring (FiPS), a novel algorithm that leverages the relationship between parameter sharing, tensor decomposition, and sparsity to efficiently compress large vision transformer models. FiPS employs a shared base and sparse factors to represent shared neurons across multi-layer perception (MLP) modules. Shared parameterization is initialized via Singular Value Decomposition (SVD) and optimized by minimizing block-wise reconstruction error. Experiments demonstrate that FiPS compresses DeiT-B and Swin-L MLPs to 25–40% of their original parameter count while maintaining accuracy within 1 percentage point of the original models.

## 1 INTRODUCTION

Over the last decade, large neural networks have achieved impressive performance across various tasks by scaling up datasets and model sizes. However, this growth has led to computational, memory, and storage challenges, necessitating efficient model compression techniques to reduce overhead and enable deployment on resource-constrained devices like mobile phones and embedded systems. To this end, research has explored various approaches, including tensor decomposition, quantization, distillation, sparsity, parameter sharing, and adaptive computing methods (Cheng et al., 2020). While most of these methods are well-studied and successfully utilized in practice (e.g., distillation, quantization), parameter sharing has received less attention.

Sharing parameters across multiple layers of a neural network could, in theory, reduce memory requirements and increase cache hits, leading to faster execution. Motivated by this, several previous works have explored reusing entire transformer blocks when defining a network (Lan et al., 2020; Takase & Kiyono, 2023; Lin et al., 2023), resulting in more efficient models. Although the ability to share weights unmodified across layers is promising, we hypothesize that a more fine-grained approach may achieve better compression, leading us to focus on sharing neurons across different layers.

We show that sharing neurons across layers can be achieved using a shared basis, where each neuron is computed as a linear combination of this basis. Crucially, we find that sparsity in the projection matrix is essential for this approach to be effective. This insight leads to our novel parameter sharing algorithm, **Fi**ne-grained **P**arameter **S**haring (FiPS), which we demonstrate effectively compresses large vision transformer models. Our contributions include:

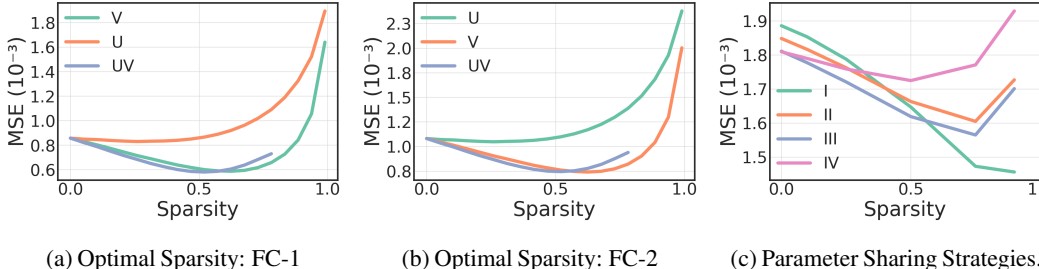

(a) Optimal Sparsity: FC-1   (b) Optimal Sparsity: FC-2   (c) Parameter Sharing Strategies.

Figure 2: **Initial Experiments.** Reconstruction error when inducing sparsity on different factors of low-rank decomposition of (a) the FC-1 layer and (b) the FC-2 layer, under a parameter budget of 25%. Higher sparsities, when applied to larger factors, enable higher ranks and achieve lower error. (c) Average reconstruction error of four FC layers under different parameter sharing strategies and sparsities. See § 2.2 for a description of each strategy.

- We demonstrate the feasibility of neuron sharing across layers by leveraging sparse tensor decomposition in MLP modules, enabling efficient parameter sharing with minimal performance degradation.
- We explore various neuron-sharing configurations by analyzing different concatenation strategies for MLP modules.
- Building on these insights, we introduce FiPS, a method that employs Singular Value Decomposition (SVD) to initialize the compressed model and optimizes the decomposed parameters by minimizing block-wise reconstruction error.
- We apply FiPS (using both unstructured and structured sparsity) to compress the MLP modules of DeiT-B (Touvron et al., 2021) and SWIN-L (Liu et al., 2021), reducing their size to 25–40% of the original while preserving accuracy with a minimal loss (under 1 percentage point).

## 2 PARAMETER SHARING THROUGH SPARSE TENSOR DECOMPOSITION

Consider a weight matrix, $\mathbf{W} \in \mathbb{R}^{d \times p}$, which projects feature vectors from $d$-dimensional space to a $p$-dimensional space, with neurons represented as the columns of $\mathbf{W}$. We aim to share weights across a subset of these $p$ neurons such that there remain only $r < p$ unique neurons; in other words, we want only $r$ columns of $\mathbf{W}$ to have unique values. We can represent the $r$ unique neurons using a lookup table (basis matrix) $\mathbf{U} \in \mathbb{R}^{d \times r}$. Then, our original matrix $\mathbf{W}$ can be reconstructed using an $r$-dimensional one-hot vector for each of its $p$ columns, represented by a projection matrix $\mathbf{V} \in \mathbb{R}^{r \times p}$. This is the "one-hot" approach illustrated in the upper part of Figure 1. Note that the number of unique neurons in this setting is $r$, limiting the resulting matrix's representation power $\mathbf{W}$. One way to alleviate this limitation is to increase the number of non-zero elements in $\mathbf{V}$, effectively creating combinations of the basis neurons and resulting in significantly more unique neuron representations, as illustrated in the lower part of Figure 1. So far, we have considered sharing neurons within a single matrix $\mathbf{W}$. However, this approach can be easily extended to multiple weight matrices $\mathbf{W}_1, ..., \mathbf{W}_N$. Specifically, we can enable fine-grained parameter sharing across multiple layers by increasing the size of our projection matrix $\mathbf{V}$ and shared basis $\mathbf{U}$.

The approach outlined above can be viewed as a low-rank decomposition of a matrix $\mathbf{W}$, where the first factor $\mathbf{U}$ is shared and the second factor $\mathbf{V}$ is sparse. Thus, we can utilize existing low-rank decomposition techniques to obtain an optimal shared orthogonal basis and induce sparsity in the projection matrices using existing pruning and sparse training techniques. In what follows, we use a pre-trained DeiT-B model (with 12 encoder blocks, each containing one MLP modules, pre-trained on ImageNet-1k (Deng et al., 2009)) and investigate the best strategy for tying multiple layers using the described framework. Specifically, we focus on the model's MLP modules, each consisting of two fully connected (FC) layers with dimensions $\mathbb{R}^{d \times p}$ and $\mathbb{R}^{p \times d}$ respectively, where $p = 4d$.

### 2.1 OPTIMAL SPARSITY FOR TENSOR DECOMPOSITION

Before moving on to parameter sharing through shared bases, we decompose individual FC layers using a truncated SVD with a 25% parameter budget and introduce sparsity by setting low-magnitude values to zero. We consider introducing sparsity in: (1) $\mathbf{U}$, (2) $\mathbf{V}$, and (3) both $\mathbf{U}$ and $\mathbf{V}$. We vary

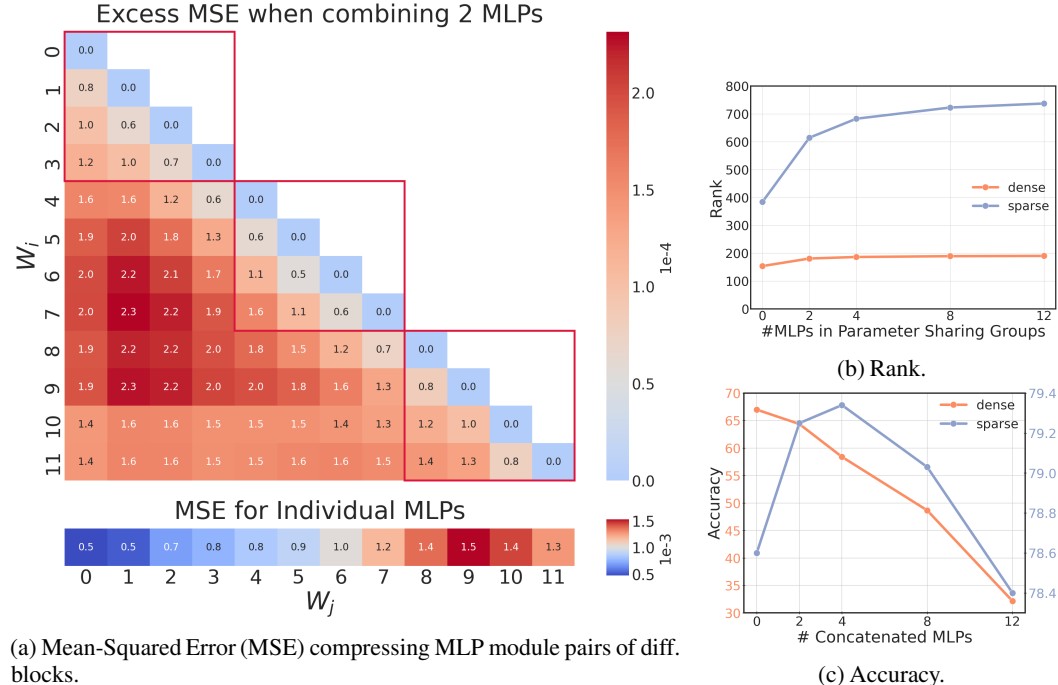

(a) Mean-Squared Error (MSE) compressing MLP module pairs of diff. blocks.

(b) Rank.

(c) Accuracy.

Figure 3: **Parameter Sharing Groups**. (a *top*) Increase in mean squared error (MSE) when sharing $\mathbf{U}$ across different MLP modules. Combining adjacent MLP modules leads to better reconstruction, as indicated by the red squares. (a *bottom*) MSE when compressing individual MLP modules, with sharing $\mathbf{U}$ among consecutive layers typically yielding the lowest error. (b) For a fixed parameter budget, the rank of the shared basis $\mathbf{U}$ stabilizes around four MLP modules. (c) This matches the optimal group size for maximizing accuracy when compressing the DeiT-B model with our algorithm.

the sparsity of matrices while keeping the total number of non-zero parameters fixed. The resulting reconstruction errors are shown in Figures 2a and 2b. We observe that the best errors are achieved around 60–80% sparsity and when sparsity is introduced on the larger factor (i.e., $\mathbf{V}$). We believe this is because larger matrices have more redundant weights and, thus, easier to prune.

## 2.2 WEIGHT CONCATENATION AND FINDING SHARED DIMENSIONS

Next, we study introducing parameter sharing across multiple layers within a network. Specifically, we consider four FC layers from two different MLP modules and concatenate their parameters in different ways to find the optimal strategy for constructing the shared basis.

First, we transpose the parameters of the second FC layers in each MLP module, such that each layer is represented by a weight matrix $\mathbf{W} \in \mathbb{R}^{d \times 4d}$. We then explore four distinct ways of concatenating 2 MLP modules together (four FC layers in total):

(I) Concatenate all matrices along the longer axis: $\mathbf{W_s} \in \mathbb{R}^{d \times 16d}$.
(II) Concatenate FC layers from the same module along the longer axis and then different modules along the shorter: $\mathbf{W_s} \in \mathbb{R}^{2d \times 8d}$.
(III) Contrary to (II), concatenate the layers from the same module along the shorter axis and then different modules along the longer: $\mathbf{W_s} \in \mathbb{R}^{2d \times 8d}$.
(IV) Concatenate all layers along the shorter axis: $\mathbf{W_s} \in \mathbb{R}^{4d \times 4d}$.

We apply truncated SVD to the concatenated matrix $\mathbf{W_s}$, and keep the top $r$ singular vectors. The resulting matrix $\mathbf{V}$ (corresponding to the right singular vectors) is sparsified by keeping the entries with the largest magnitude, an approach shown in § 2.1 to yield optimal reconstruction. Finally, we reconstruct the parameters using a shared basis and report the mean squared error (MSE) in Figure 2c. Concatenating weights along the longer dimension yields the best reconstruction error, especially

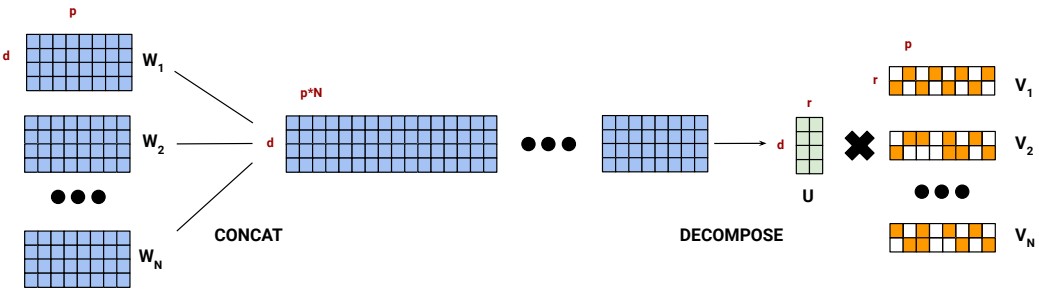

Figure 4: **Parameter Sharing Through Sparse Tensor Decomposition.** A group of FC layers are concatenated along the larger dimension, $p$, and decomposed into two matrices: a shared basis, $\mathbf{U}$, and a sparse projection matrix, which is then sliced up respectively for each layer.

at higher sparsity levels. Therefore, going forward, we will always concatenate the two FC layers of every MLP module in the transformer blocks along their larger dimension.

### 2.3 PARAMETER SHARING ACROSS LAYERS

In this section, we study the redundancy and interplay among different MLP modules to identify ideal groups for parameter sharing. First, we decompose individual modules using a rank of $r = 180$ and plot the MSE in Figure 3a-bottom. We observe that the error increases almost monotonically with the module index, suggesting the need for allocating more capacity to later modules in the network.

Next, we group two MLP modules from two blocks ($i$ and $j$) and make them share the same basis $\mathbf{U}$, which reduces the parameter count and consequently increases the MSE for each block. The MSE for block $i$ in this shared setting is denoted as $MSE_{i,j}$. In Figure 3a, we plot the average MSE increase between blocks $i$ and $j$ as $MSE_{i,j}^{\uparrow} = (MSE_i - MSE_{i,j} + MSE_j - MSE_{j,i})/2$. Empirically, we find that nearby blocks tend to have the smallest increase in error, motivating the grouping of consecutive layers when sharing parameters.

We then study the optimal number of MLP modules per group. Increasing the group size, thereby sharing more parameters across the same basis $\mathbf{U}$, allows for a higher rank, as shown in Figure 3b. This effect is more pronounced when sparsity is applied to factor $\mathbf{V}$. However, a higher rank does not always lead to better accuracy, as the shared basis must cover a larger number of parameters. As illustrated in Figure 3c, the highest post-compression accuracy is attained when parameters are shared across the MLP modules four consecutive blocks[†].

## 3 SPARSITY-ENABLED PARAMETER SHARING

Experiments in the previous section motivate and guide us in developing FiPS, an efficient parameter-sharing algorithm enabled by sparse tensor decompositions that can be summarized in three main points:

1. **Shared Initialization:** We tie multiple FC layers across a group of MLP modules and apply low-rank decomposition via truncated SVD.
2. **Local Error Minimization:** We finetune our shared low-rank initialization to minimize the difference between the activations of the original and compressed model. During this step, we also introduce sparsity in our factors, which helps us allocate parameters where they are most needed.
3. **Global Error Minimization (Optional):** For best results, especially at lower compression levels, we finetune our compressed models end-to-end.

Algorithm 1 outlines the key steps of FiPS, which are detailed below.

**Shared Initialization.**   We begin by compressing the pre-trained model through parameter sharing, achieved by concatenating and decomposing multiple FC layers simultaneously, as illustrated in Fig-

---

[†]Here, we compress the DeiT-B architecture using FiPS, which is introduced in the following section.

ure 4. For higher parameter budgets and sparsity levels (e.g., 50% and 75%, respectively), the rank of our low-rank factor $\mathbf{U}$ can exceed the model dimension $d$. In such cases, we grow the matrices $\mathbf{U}$ and $\mathbf{V}$ similar to the approach in Net2Net (Chen et al., 2016). However, unlike Net2Net, we grow $\mathbf{U}$ by appending zeros rather than splitting each neuron. We select the top-$k$ neurons with the highest singular values (i.e., $k = r - d$) and multiply them by $1/\tau$, where $\tau$ is treated as a hyperparameter (discussed further in § 5).

Formally, the parameters of a group of FC layers, $\mathbf{W}_1, \mathbf{W}_2, ..., \mathbf{W}_N$, are concatenated into a large matrix $\mathbf{W_s} = [\mathbf{W}_1; \mathbf{W}_2; ...; \mathbf{W}_N]$, where $\mathbf{W}_i \in \mathbb{R}^{d \times p}$ [†]. We then apply truncated SVD, $\mathbf{W_s} = \mathbf{U} \mathbf{\Sigma} \hat{\mathbf{V}}$, to obtain a low-rank approximation of the parameters, where $\mathbf{U} \in \mathbb{R}^{d \times r}$, $\mathbf{\Sigma} \in \mathbb{R}^{r \times r}$, and $\hat{\mathbf{V}} \in \mathbb{R}^{r \times (N \cdot p)}$. The factor $\mathbf{U}$ is shared among all layers within the group and remains dense due to its relatively small size. Next, we multiply $\hat{\mathbf{V}}$ by the singular values to obtain the projection matrix $\mathbf{V} = \mathbf{\Sigma} \hat{\mathbf{V}}$. Finally, the weights are reconstructed as $\mathbf{W}'_i = \mathbf{U} \mathbf{V}_i$, where each $\mathbf{V}_i$ is a slice of $\mathbf{V}$ corresponding to the weight matrix $\mathbf{W}_i$.

**Local Error Minimization**   For the second phase of our method, we compute the input and output activations of the original FC layers using a calibration dataset $D$, described in § 4. We use these activations to optimize the compressed layers and minimize the *L2-loss* between the original and compressed layers' activations:

$$\underset{\mathbf{U}, \mathbf{V_i}, ..., \mathbf{V_N}}{\arg\min} \sum_i^N \|\mathbf{W}_i \mathbf{X}_i - \mathbf{U} \mathbf{V}_i \mathbf{X}_i\|_2^2, \tag{1}$$

where $\mathbf{X}_i$ is the inputs to the $i^{\text{th}}$ original FC layer. We explore several sparse training and pruning techniques to identify a sparse $\mathbf{V}$ during this optimization: (a) *Static Sparsity*, which establishes the sparsity structure by retaining the top-magnitude connections before training (Hoefler et al., 2021); (b) *Gradual Magnitude Pruning (GMP)* (Zhu & Gupta, 2017), which progressively increases sparsity by updating its mask every $T$ steps, retaining the top-magnitude connections following the cubic schedule from Kurtic et al. (2023); and (c) *RigL* (Evci et al., 2021), which starts from (a) but updates the sparse connectivity every $\Delta T$ steps using gradient and magnitude information. We decided to use *GMP* for the final sparse training recipe due to its superior performance.

Although we share parameters across multiple MLP modules, gradients for error minimization can be computed one MLP module at a time. Therefore, optimization requires significantly fewer resources compared to end-to-end fine-tuning.

**Global Error Minimization.**    In this optional stage, we finetune the shared parameterization found in the previous stage end-to-end to further improve our results. Because our factors $\mathbf{V}_i$ are sparse, we employ the dynamic sparse training method, *RigL*, during this stage as we observe it to perform slightly better than keeping the sparsity pattern constant (i.e., *Static Sparsity*) as discussed in § 4.

**Latency and Memory profiling**   As noted in § 3, for 75% sparsity, the parameter budgets above 27.5% increase the rank of the shared singular vectors *beyond* the original model embedding dimension. We require efficient sparse operations and representations to effectively reduce the latency and memory overhead of FiPS.

Implementing dedicated kernels to fully exploit the potential for compression offered by FiPS is outside the scope of this work; however, we demonstrate promising preliminary benchmarks by utilizing NVIDIA's tensor core support for 2:4 (Mishra et al., 2021) sparsity for GPUs, and Neural Magic's DeepSparse Engine (Neural Magic, 2021) to showcase CPU performance. See fig. 8 for latency and memory overhead comparisons of DeiT-B. Note that the parameter budgets are slightly adjusted for latency benchmarking to ensure tensor shapes are evenly divisible by 64, a necessary property to leverage 2:4 sparsity with 16-bit data types. Based on the latency profiling results in fig. 8 and the structured sparsity performance in table 3, FiPS demonstrates effective model compression, offering improvements in both memory usage and latency.

## 4 MAIN RESULTS

**Experimental Setup**   In our experiments, we used DeiT-B (with 12 blocks) and Swin-L (four stages containing 2, 2, 18, and 2 blocks, respectively) (Touvron et al., 2021; Liu et al., 2021). We used a calibration dataset $D$ of $30 \times 128 = 3840$ samples from ImageNet-1k (Deng et al., 2009). We found that 20

---

[†] We transpose the second FC layer's parameters to match the shape of the first FC layer.

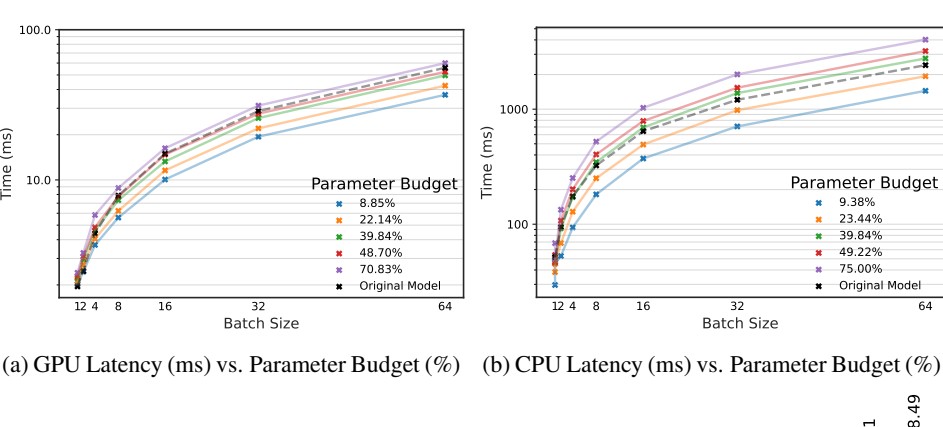

(a) GPU Latency (ms) vs. Parameter Budget (%)     (b) CPU Latency (ms) vs. Parameter Budget (%)

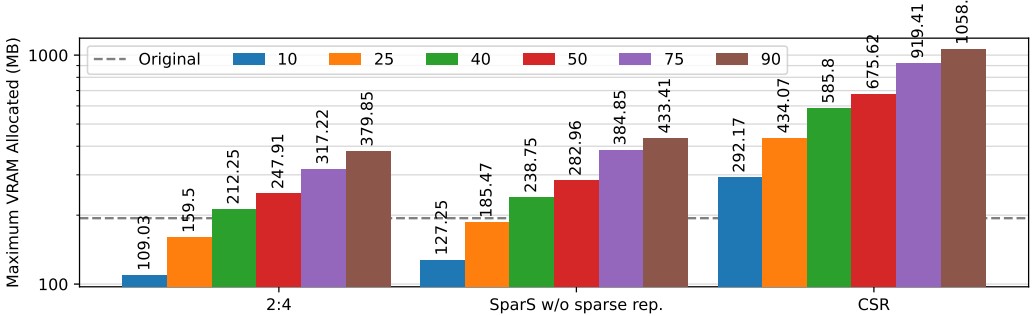

(c) VRAM (MB) vs. compression strategy for FiPS at various parameter budgets (%) at batch size 64

Figure 5: **DeiT-B inference latency and memory benchmarks.** (a) End-to-end latency of 2:4 sparse FiPS on an Nvidia A4000 for batch sizes ranging from 1 to 64. FiPS with a 22% parameter budget exhibits a 25% latency improvement over the original network above batch sizes of 8. (b) Latency of 75% unstructured sparse FiPS accelerated with DeepSparse Engine on Intel Xeon W-2145 CPU. On CPU, FiPS with a 23% parameter budget is faster than the original network at all batch sizes measured. (c) Maximum VRAM allocation for 50% sparse FiPS using 2:4, strided (i.e., without a sparse representation), and CSR tensor storage. At 10 and 25% parameter budgets, 2:4 reduces maximum allocated memory of 18 to 44%, respectively. CSR increases memory overhead at this modest sparsity due to the associated overhead of storing the non-zero element indices. We find that the reduction in memory overhead is consistent for all batch sizes observed from 1 to 64. Note that all plots in fig. 8 have a logarithmic y-axis.

---

**Algorithm 1** **Fi**ne-grained **P**arameter **S**haring

---

**Require:** MLP parameters $\mathbf{W}_1, \cdots, \mathbf{W}_N \in \mathbb{R}^{d \times p}$, MLP inputs $\mathbf{A}_i$ and MLP function $\mathbf{f}(\mathbf{W}_i, \mathbf{A}_i)$,
     Target rank $r$, Learning Rate $\eta$, Steps $\mathbf{T}$.
1:   $\mathbf{U}, [\mathbf{V}_1, \mathbf{V}_2, \cdots, \mathbf{V}_N] \leftarrow TruncatedSVD([\mathbf{W}_1; \mathbf{W}_2; \cdots; \mathbf{W}_N], k = r)$
2: **for** each training iteration $t = 1$ to $T$ **do**
3:      $\mathbf{G}_{\mathbf{U}} = 0$                                                    ▷ Gradient accumulator for $\mathbf{U}$
4:      **for** each block $i$ **do**
5:          $\mathbf{V}_i \leftarrow Sparsify(\mathbf{V}_i, t)$                      ▷ Potentially increase or adjust sparsity
6:          $L_i \leftarrow MSE\_loss(\mathbf{f}(\mathbf{W}_i, \mathbf{A}_i), \quad \mathbf{f}(\mathbf{U}\mathbf{V}_i, \mathbf{A}_i))$
7:          $\mathbf{V}_i \leftarrow \mathbf{V}_i - \eta \nabla_{\mathbf{V}_i} L_i$
8:          $\mathbf{G}_{\mathbf{U}} \leftarrow \mathbf{G}_{\mathbf{U}} + \nabla_{\mathbf{U}} L_i$
9:      **end for**
10:     $\mathbf{U} \leftarrow \mathbf{U} - \dfrac{\eta}{N} \mathbf{G}_{\mathbf{U}}$
11: **end for**
12: **return** $\mathbf{U}, [\mathbf{V}_1, ..., \mathbf{V}_N]$

---

Table 1: **Compressing Models Pretrained on ImageNet-1k**. We compare top-1 val. accuracy of DeiT-B (81.85%) (Touvron et al., 2021) and Swin-L (86.24%) (Liu et al., 2021) models compressed using FiPS vs. AAFM/GFM across parameter budgets. Results include layer-wise error minimization (FiPS) and global error minimization (FiPS + FT). AAFM/GFM[†] results are from Yu & Wu (2023).

| Param. Budget | 10% | | 25% | | 40% | | 50% | | 75% | |
|---|---|---|---|---|---|---|---|---|---|---|
| Method / Model | DeiT | Swin | DeiT | Swin | DeiT | Swin | DeiT | Swin | DeiT | Swin |
| AAFM [†] | – | – | – | – | 80.33 | – | 81.21 | 85.04 | – | – |
| GFM [†] | – | – | – | – | 81.28 | – | 81.62 | 85.33 | – | – |
| FiPS (ours) | 70.04 | 74.04 | 80.64 | 84.78 | **81.69** | **85.69** | 81.83 | 85.99 | 81.82 | 86.21 |
| FiPS + FT (ours) | **77.26** | **82.13** | 81.31 | 85.16 | 81.55 | 85.68 | 81.54 | **85.99** | – | – |

Table 2: **Transfer Learning Results**. We compare the Top-1 accuracy of the original DeiT-B model with compressed versions using GFM and FiPS across various MLP parameter budgets on small-scale datasets. GFM[†] and Original[†] results are from Yu & Wu (2023) and Touvron et al. (2021), respectively.

| Models | Original[†] | GFM[†] | | FiPS+RigL FT (ours) | | |
|---|---|---|---|---|---|---|
| Param. Budget | 100% | 40% | 50% | 25% | 40% | 50% |
| CIFAR-100 | 90.99 | 90.17 | 90.67 | 90.88 | 91.24 | **91.33** |
| Pets | **94.74** | 93.95 | 94.22 | 94.19 | 94.52 | 94.41 |
| Flowers102 | 97.77 | 97.02 | 97.45 | 97.84 | 98.14 | **98.37** |
| iNaturalist-2019 | 77.39 | 77.13 | 77.56 | 77.26 | 77.58 | **77.69** |

epochs with this calibration set yielded near-optimal results, while longer training or larger calibration sets yielded marginal improvements. This calibration stage took <1 hour on an Nvidia A6000 GPU for both architectures. Transfer learning capability was assessed on CIFAR-100, Flowers102, Oxford-III-Pets, and iNaturalist 2019 datasets (Krizhevsky, 2009; Nilsback & Zisserman, 2008; Parkhi et al., 2012; Van Horn et al., 2018) using 100 training epochs, following the methodology of Yu & Wu (2023). In all our experiments, we used *AdamW* optimizer (Loshchilov & Hutter, 2019) and identified the optimal learning rates by performing a small hyperparameter sweep using 12 logarithmically spaced values.

When applying FiPS, we targeted 75% average sparsity across all sparse factors, as this resulted in the best compression, as shown in Figure 6b. The mask update interval, $\Delta T$, for both *RigL* and *GMP* was set to 50 steps. When using *RigL* during global fine-tuning, we used an initial pruning ratio[†] of $0.1$ and reduced this value to $0.05$ during our transfer learning experiments to limit changes in the sparsity pattern. Further hyperparameter details for the optimizer and the sparse training algorithms used are provided in Appendix A.2.

Following our earlier results (i.e., Figure 3c), we used groups of four consecutive blocks (each with one MLP module) for the DeiT-B architecture, resulting in three different parameter-sharing groups. For Swin, we shared parameters within each 2-block stage separately and split the remaining stage with 18 blocks into three groups with six consecutive blocks each.

**ImageNet-1k** We compare FiPS against the baselines of Adaptive Atomic Feature Mimicking (AAFM), which utilizes block-wise error minimization, and Global Feature Mimicking (GFM), combining AAFM with distillation at the model output, both of which are proposed by Yu & Wu (2023). At a 40% parameter budget, FiPS achieves 1.36% point higher accuracy (shown in Table 1) than AAFM and even exceeds the costly GFM approach by 0.41% point, which requires significantly higher memory and computing resources due to the need for end-to-end fine-tuning.

As for Swin-L, the picture is similar as shown in Table 1. Across all parameter budgets, *GMP*-based FiPS consistently achieves higher accuracies than alternatives like GFM while requiring less computing and memory to compress the pre-trained model.

---

[†]Pruning ratio is the proportion of non-zero elements pruned and regrown at each mask update step.

**Transfer Learning** Next, we take our compressed models and finetune them on four different transfer tasks. Since the factors are already sparse, we use RigL to adapt the sparse factors. Models compressed through FiPS result in significantly better transfer accuracies as shown on Table 2.

## 5 ABLATIONS

In this section, we study the importance of various components of the FiPS algorithm when compressing the DeiT-B model at different parameter budgets. First, we ablate key components of our algorithm in Figure 6a under a 25% parameter budget:

1. **SVD-based Initialization:** Using a random initialization (RI) instead of SVD-based initialization results in a 1% point drop in accuracy.

2. **Global Pruning:** We use global pruning (GP) when sparsifying our sparse factors $\mathbf{V}$, which results in 0.4% point improvement over local pruning (LP), which enforces the same sparsity level for each factor $\mathbf{V}_i$.

3. **Scaling Vectors:** We normalize the weights $\mathbf{W}$ before the initial SVD stage of FiPS as suggested by Liu et al. (2024). Normalized weights initialize the SVD, while magnitude vectors initialize scaling vectors (SV) for neuron scaling. Scaling vectors improve local pruning but are less effective with global pruning.

Consequently, the final FiPS configuration integrates GMP with GP. Next, we perform a sensitivity analysis using different sparsity levels, calibration dataset sizes, and training lengths using the DeiT-B checkpoint trained on ImageNet-1k.

**Optimal Sparsity for Sparse Factors** We compressed the DeiT-B model, as described in § 4, using sparsity levels ranging from 50% to 96% at a reduced parameter budget of 17.5%, as shown in Figure 6b. The best performance was observed at 75% sparsity as shown in Figure 6b. While increasing sparsity to 87% yielded similar accuracy, lowering it to 50% resulted in a notable drop in performance, likely due to a significant reduction in rank.

**Calibration Dataset Size and Training Length** We examine the relationship between the number of batches and epochs using a fixed batch size of 128 and a 50% parameter budget. We require the calibration set to have at least 3 data points per category for generalization (i.e., 30 batches) and observe that increasing the number of calibration data points results in less than 0.1% point improvement. Similarly, training beyond 20 epochs often results in worse generalization.

**Alternative Sparsity Techniques** In addition to *GMP*, we considered using *Dense* tensor decompositions (i.e., no sparsity on $\mathbf{V}$ factors) and other sparse training techniques: *Static Sparsity* and *RigL*. Results are presented in Table 4. In the case of DeiT-B, for parameter budgets from 10–50%, *RigL* consistently outperforms *Dense* and *Static Sparsity*. At higher budgets, all methods converge to nearly identical accuracies approaching the original model accuracy. For Swin-L, *RigL* outperforms *Dense* and *Static Sparsity* at 10% and 25% parameter budgets. However, for higher parameter budgets, *Static Sparsity* obtains slightly higher accuracies.

**Structured Sparsity** We investigate the generalization performance of FiPS using structured sparsity, with results presented in table 3. The methods explored include the Straight Through Estimator (STE) employing top-k weight magnitude selection, projects parameters into a sparse subspace during training, and applies gradients to dense parameters with a gradual pruning schedule for improved results; the Sparse-Refined Straight Through Estimator (SR-STE), which mitigates the negative impact of approximated gradients; and $N:M$ Structured RigL (NMSRigL) and $N:M$ Structured GMP (NMSGMP) (Lee et al., 2023; Zhou et al., 2021; Lasby et al., 2024), where $N:M$ denotes the sparse weight matrix dimensionality (e.g., a 50% sparsity in FC weight matrices of size $d \times 4d$ corresponds to a 2:4 structure).

**Sparsity Distribution and MSE-loss** Figure 7a shows that earlier MLP layers are easier to compress, requiring fewer parameters, while later layers are more challenging, as reflected by higher reconstruction errors in Figure 3a. These later layers exhibit lower sparsity and higher weight density, with Figure 7b highlighting a strong correlation (0.922) between weight density and MSE. This indicates that later layers demand more parameters to preserve performance.

Table 3: **Structured Sparsity Performance**. ImageNet top-1 validation accuracy (%) of DeiT-B (81.85%) (Touvron et al., 2021) for various structured sparsification methods at 50% and 75% sparsity, compared to Unstructured FiPS. Methods include STE, Sparse-Refined STE, $N : M$ Structured RigL (NMSRigL), and $N : M$ Structured GMP (NMSGMP) at 50% sparsity, corresponding to 2:4 structures (Lee et al., 2023; Zhou et al., 2021; Lasby et al., 2024).

| Param. Budget (at 50% Sparsity) | 10% | 25% | 40% | 50% |
|---|---|---|---|---|
| STE | 42.89 | 73.26 | 78.26 | 79.36 |
| SR-STE | 45.31 | 75.53 | 79.71 | 80.68 |
| NMSRigL | 44.87 | 75.71 | 79.97 | 80.99 |
| NMSGMP | 52.36 | 76.88 | 80.59 | 81.31 |
| Unstructured FiPS | 54.00 | 77.56 | 80.94 | 81.63 |
| Unstructured FiPS (75% Sparsity, optimal) | 70.04 | 80.64 | 81.69 | 81.83 |

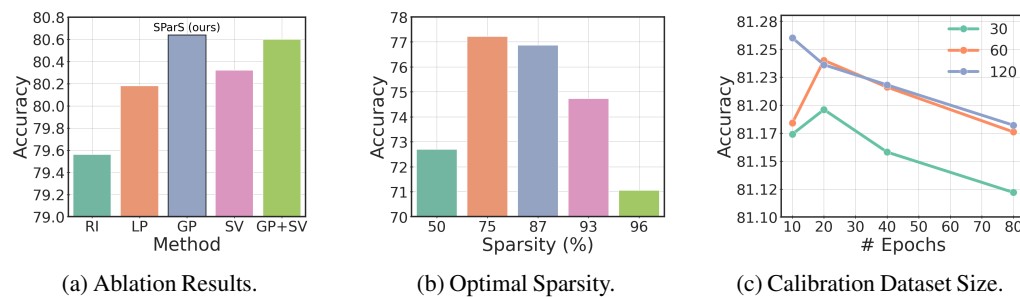

(a) Ablation Results.  (b) Optimal Sparsity.  (c) Calibration Dataset Size.

Figure 6: **Ablations and Sensitivity Analysis.** ((a) Ablation of key ingredients of the FiPS algorithm: random initialization (RI), Local Pruning (LP), Global Pruning (GP), and Scaling Vectors (SV); (b) Effect of sparsity used in sparse factors; (c) Effect of calibration dataset size (denoted as #batches with 128 images each) and # training epochs to post-compression accuracy.

Table 4: **Sparsification Method and FiPS Generalization Performance**. ImageNet top-1 validation accuracy (%) of DeiT-B (81.85%) (Touvron et al., 2021) and Swin-L (86.24%) (Liu et al., 2021) models compressed with FiPS using different sparsity methods: RigL (Evci et al., 2021) and static sparsity.

| Param. Budget | 10% | | 25% | | 40% | | 50% | | 75% | |
|---|---|---|---|---|---|---|---|---|---|---|
| Method / Model | DeiT | Swin | DeiT | Swin | DeiT | Swin | DeiT | Swin | DeiT | Swin |
| Dense | 15.35 | 3.61 | 65.71 | 60.31 | 74.33 | 80.61 | 79.22 | 83.59 | 81.36 | 85.64 |
| Static Sparsity | 65.26 | 65.6 | 80.06 | 84.37 | 81.48 | **85.69** | 81.70 | 85.98 | **81.86** | **86.23** |
| RigL | 66.67 | 70.96 | 80.31 | 84.57 | 81.50 | 85.59 | 81.65 | 85.91 | 81.82 | 86.20 |
| GMP (FiPS) | **70.04** | **74.04** | **80.64** | **84.78** | **81.69** | 85.69 | **81.83** | **85.99** | 81.82 | 86.21 |

**Growing Neurons in Shared Bases and Sparse Factors**   As discussed in § 3, high parameter budgets and sparsity levels (e.g., 50% and 75% for DeiT-B) often allow the rank $r$ to exceed the model dimension $d$. Since SVD provides only $d$ directions for initialization, we explore three methods for initializing the remaining $k = r - d$ dimensions: 1. **Random Growth:** New neurons in $\mathbf{U}$ are initialized to zero, and those in $\mathbf{V}$ are initialized randomly using He et al. (2015); 2. **Neuron Splitting:** The top $k$ neurons of $\mathbf{U}$ are duplicated, and the top $k$ neurons of $\mathbf{V}$ are halved, following Chen et al. (2016); 3. **Hybrid Initialization:** New neurons in $\mathbf{U}$ are initialized to zero, while those in $\mathbf{V}$ are derived from the top $k$ neurons and normalized by $\tau$. This approach minimizes the immediate impact of new neurons in $\mathbf{V}$, allowing optimization to gradually re-activate them, as suggested by Evci et al. (2022). After a hyperparameter sweep on $\tau$, the hybrid initialization outperformed the alternatives, achieving 1% and 2% higher accuracy than methods (1) and (2), respectively.

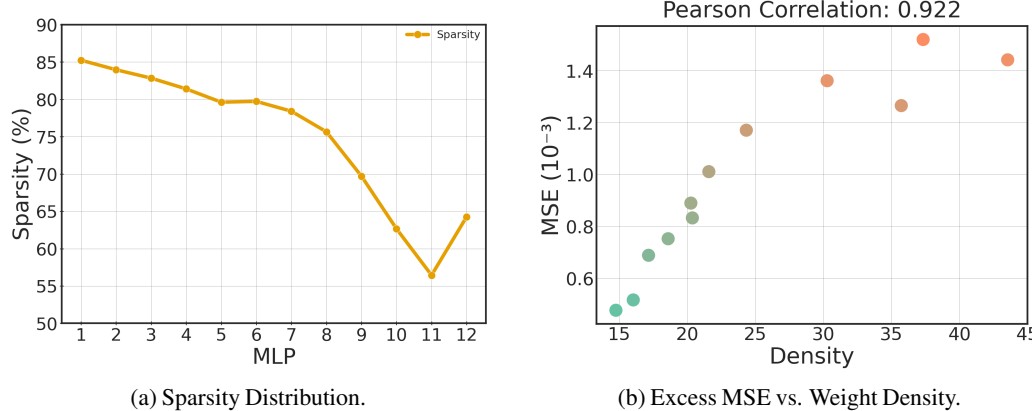

(a) Sparsity Distribution.

(b) Excess MSE vs. Weight Density.

Figure 7: **Effectiveness of Global Sparsity.** (a) Average sparsity in the MLP modules of DeiT-B at the end of the training, showing more parameters are allocated to later modules. (b) We observe a high correlation between the MSE values reported in Figure 3a and the parameter distribution found by FiPS.

## 6 RELATED WORK

**Vision Transformers (ViT)** were introduced by Dosovitskiy et al. (2021) for image recognition using image patches as sequences, similar to tokens in NLP models. Touvron et al. (2021) improved ViT's data efficiency by introducing a distillation token during training. Liu et al. (2021) proposed Swin Transformers, which use a hierarchical structure and shifted windows for local self-attention, differing from ViT and DeiT.

**Sparsity in Neural Networks** includes three main approaches: post-training pruning, sparsifying during training, and fully sparse training (Hoefler et al., 2021). Early methods involved heuristic pruning, such as removing the smallest magnitude parameters (Thimm & Fiesler, 1995). Later approaches, like GMP (Zhu & Gupta, 2017), increased the amount of pruning, while dynamic pruning with accelerated schedulers was explored by Kurtic et al. (2023). Static sparsity uses a pre-initialized mask throughout training (Hoefler et al., 2021), whereas dynamic methods, like RigL (Evci et al., 2021), adjust the sparsity pattern during training based on gradient information.

**Tensor Decomposition** techniques, such as those presented by Kolda & Bader (2009), reduce redundancy in large weight matrices using low-rank decomposition. Yu & Wu (2023) introduced AAFM for transformer models, using truncated PCA to reconstruct weights, and GFM to minimize loss between compressed and original models, similar to Knowledge Distillation (Hinton et al., 2015).

**Parameter Sharing** is less explored but includes works like Eban et al. (2019), who used a Sum-Product reducer to map shared parameters, and Obukhov et al. (2021), who employed TR decomposition for shared parameters in 3D tensors. Zhang et al. (2022) introduced "Weight Multiplexing," sharing parameters between MLP modules in ViT, alongside distillation and linear projections between transformer blocks to aid model recovery.

## 7 CONCLUSION

This work introduces FiPS, demonstrating for the first time that inter-layer parameter sharing enables significant compression in Transformers. While this study focuses on ViT backbones and MLP modules, similar gains are expected with multi-head attention parameters, leading to greater compression. Further improvements may be possible through quantization of the current full-precision bases, which we leave for future work.

AUTHOR CONTRIBUTIONS

Cem Üyük led the project, proposed and executed the experimental plan, facilitated the team meetings, developed the majority of the software architecture, implemented static sparse training and provided code review for the sparse training algorithms, wrote the initial draft of the paper, and continued con-

tributing to writing significantly while also creating most of the plots. Mike Lasby implemented sparse training algorithms, assisted the software architecture development, handled distributed training integration, performed code reviews, and assisted with writing and proofreading the paper. Mohamed Yassin assisted with coding and running inference experiments. Utku Evci proposed the project and its central idea, contributed to the research plan and direction, advised Cem, reviewed the code, helped substantially with the writing, and created some of the plots. Yani Ioannou helped with the research direction, contributed to the paper's motivation, helped with the writing, provided compute resources, and supervised the work by members of the Calgary ML Lab at the University of Calgary, including Cem Üyük (Visiting Student Researcher), Mike Lasby (PhD Student), and Mohamed Yassin (Research Assistant).

## ACKNOWLEDGMENTS

We acknowledge the support of Alberta Innovates (ALLRP-577350-22, ALLRP-222301502), the Natural Sciences and Engineering Research Council of Canada (RGPIN-2022-03120, DGECR-2022-00358), and Defence Research and Development Canada (DGDND-2022-03120). This research was enabled in part by support provided by the Digital Research Alliance of Canada (alliancecan.ca) and Google Cloud. We also acknowledge Erik Schultheis' very helpful feedback with regard to custom kernel design.

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

# A APPENDIX

## A.1 MLP CONCATENATION STRATEGIES

In reference to § 2.2, below is a detailed account of MLP concatenation setups. Considering two MLPs, therefore, 4 FC layers and assuming each weight matrix $\mathbf{W}_{ij} \in \mathbb{R}^{d \times p}$ (where $p = 4d$ and $i,j \in \{1,2\}$), we identify four distinct methods for concatenating the FC weights:

I. All weights are combined along the longer dimension, resulting in:
$$\mathbf{W} = [\mathbf{W}_1; \mathbf{W}_2; \mathbf{W}_3; \mathbf{W}_4] \in \mathbb{R}^{d \times 16d}.$$

II. The FC1 and FC2 weights from each MLP are concatenated along the longer axis, followed by concatenation along the shorter axis. Specifically, we have:
$$\mathbf{W}_1 = [\mathbf{W}_{11}; \mathbf{W}_{12}] \in \mathbb{R}^{d \times 8*d} \quad \text{and} \quad \mathbf{W}_2 = [\mathbf{W}_{21}; \mathbf{W}_{22}] \in \mathbb{R}^{d \times 8d},$$
leading to:
$$\mathbf{W} = [\mathbf{W}_1; \mathbf{W}_2] \in \mathbb{R}^{2d \times 8d}.$$

III. The FC1 weights from both MLPs are concatenated along the longer axis, followed by concatenating the FC2 weights similarly. This results in:
$$\mathbf{W}_1 = [\mathbf{W}_{11}; \mathbf{W}_{21}] \in \mathbb{R}^{d \times 8d} \quad \text{and} \quad \mathbf{W}_2 = [\mathbf{W}_{12}; \mathbf{W}_{22}] \in \mathbb{R}^{d \times 8d},$$
yielding:
$$\mathbf{W} = [\mathbf{W}_1; \mathbf{W}_2] \in \mathbb{R}^{2d \times 8d}.$$

IV. Finally, all weights are concatenated along the shorter axis:
$$\mathbf{W} = [\mathbf{W}_1; \mathbf{W}_2; \mathbf{W}_3; \mathbf{W}_4] \in \mathbb{R}^{4d \times 4d}.$$

These configurations are explored empirically in Figure 2c and discussed in full in § 2.2.

## A.2 HYPER-PARAMETERS

### A.2.1 OPTIMIZER

To minimize local error, we employ a logarithmic grid for hyperparameter tuning. The learning rates for Dense, Static Sparsity, GMP, and RigL are set as follows for both DeiT-B and Swin-L:

1. Dense: $1.25 \times 10^{-4}$,

2. Static Sparsity: $2.5 \times 10^{-4}$,

3. GMP: $1 \times 10^{-3}$,

4. RigL: $1 \times 10^{-3}$.

For transfer learning, we use a linear grid, as some hyperparameters are derived from the codebase of DeiT. The optimal learning rates for FiPS are:

1. CIFAR-100: $2.5 \times 10^{-5}$,

2. Flowers102: $1 \times 10^{-4}$,

3. Oxford-III-Pets: $7.5 \times 10^{-6}$,

4. iNaturalist 2019: $1 \times 10^{-4}$.

### A.2.2 SPARSIFIER

**Global Mask Pruning (GMP)**   GMP begins with an initial sparsity level of 25%. During the training process, the sparsity is gradually increased to 50% at the 25% training mark and ultimately reaches 75% sparsity by the end of the training. The $\Delta T$ of 50 is used for update steps.

**RigL**   RigL employs an initialization phase that combines pruning with a growth ratio of 0.1 for block-wise error minimization and a growth ratio of 0.05 for transfer learning tasks with $\Delta T$ of 50 for growth and pruning ratio. This more conservative growth ratio in transfer learning helps preserve the mask obtained during the error minimization process, ensuring that important masks learned during the initial training are not lost.

### A.3   LATENCY AND MEMORY PROFILING

As noted in § 3, for 75% sparsity parameter budgets above 27.5% increase the rank of the shared singular vectors *beyond* the original model embedding dimension. To effectively reduce the latency and memory overhead of FiPS, we require efficient sparse operations and representations.

Implementing dedicated kernels to fully exploit the potential for compression offered by FiPS is outside the scope of this work; however, we demonstrate promising preliminary benchmarks by utilizing Nvidia's tensor core support for 2:4 (Mishra et al., 2021) sparsity for GPUs, and Neural Magic's DeepSparse Engine (Neural Magic, 2021) to showcase CPU performance. See fig. 8 for latency and memory overhead comparisons of Deit-B. Note that the parameter budgets are slightly adjusted for latency benchmarking to ensure tensor shapes are evenly divisible by 64, a necessary property to leverage 2:4 sparsity with 16-bit data types.

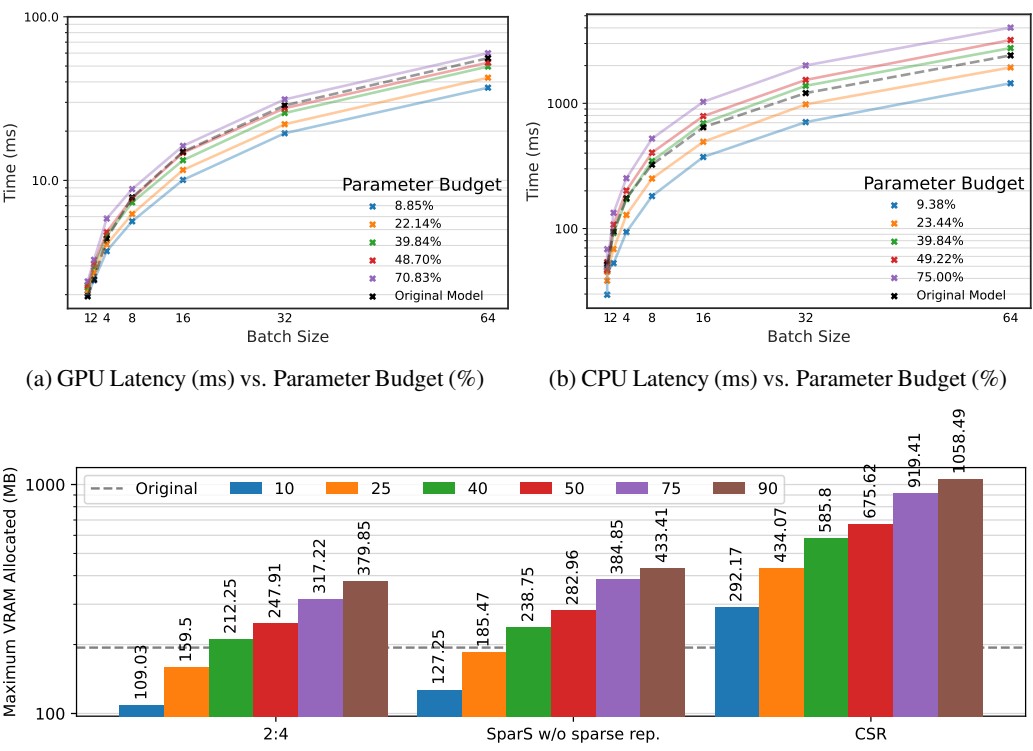

(a) GPU Latency (ms) vs. Parameter Budget (%)   (b) CPU Latency (ms) vs. Parameter Budget (%)

(c) VRAM (MB) vs. compression strategy for FiPS at various parameter budgets (%) at batch size 64

Figure 8: **DeiT-B inference latency and memory benchmarks.** (a) End-to-end latency of 2:4 sparse FiPS on an Nvidia A4000 for batch sizes ranging from 1 to 64. FiPS with a 22% parameter budget exhibits a 25% latency improvement over the original network above batch sizes of 8. (b) Latency of 75% unstructured sparse FiPS accelerated with DeepSparse Engine on Intel Xeon W-2145 CPU. On CPU, FiPS with a 23% parameter budget is faster than the original network at all batch sizes measured. (c) Maximum VRAM allocation for 50% sparse FiPS using 2:4, strided (i.e., without a sparse representation), and CSR tensor storage. At 10 and 25% parameter budgets, 2:4 reduces maximum allocated memory of 18 to 44%, respectively. CSR increases memory overhead at this modest sparsity due to the associated overhead of storing the non-zero element indices. We find that the reduction in memory overhead is consistent for all batch sizes observed from 1 to 64. Note that all plots in fig. 8 have a logarithmic y-axis.

