# OpenReview forum: "Learning Parameter Sharing with Tensor Decompositions and Sparsity"
_ICLR.cc/2025/Conference — Submitted to ICLR 2025_

### Official Review · Reviewer_Ntka · 2024-10-30

**Soundness:** 3
**Presentation:** 3
**Contribution:** 3
**Rating:** 6
**Confidence:** 3

**Summary:**

This paper introduces Sparsity-enabled Parameter Sharing (SParS), a compression algorithm for neural network parameters. This method typically focuses on compressing MLP layers in vision transformers by using shared base and sparse factors across multiple layers. Authors provide visualization and analysis on stages of the algorithm and ablate different ways of truncating MLP parameters. The authors demonstrate that their approach can compress DeiT-B and Swin-L MLPs to 25-40% of their original size while maintaining accuracy within 1% of the original models. Experiments also show that SParS (+FT) outperforms baseline methods like GFM.

**Strengths:**

- **Novel approach**: In general, the method is new and it's a smart application of sparsity on modern neural networks
- **Clear presentation**: Authors present the motivation, methodology, experiments and analysis clearly.
- **Empirical justification**: Authors use solid experiments to justify their choice of the design, i.e. on truncation strategy, number of grouped layers.
- **Strong results**: 25-40% parameter saving is in general a decent number in model compression related works.

**Weaknesses:**

- **Limited comparison**: in the main results section, authors provide few comparison with other related parameter saving methods. The limited comparative analysis makes it difficult to fully assess the method's effectiveness relative to the current state-of-the-art approaches.
- **Insufficient theoretical analysis**: while the empirical results are promising, the paper lacks sufficient theoretical analysis to explain why SParS outperforms baseline methods. A more detailed discussion of the underlying mechanisms and intuitive explanations would enhance the reader's understanding and make the technical contributions more accessible.
- **Narrow application scope**: this work mainly focuses on vision transformers, whereas parameter-efficient methods are particularly crucial for large language models. The authors should consider extending their analysis to language models or discuss potential challenges and modifications needed for such applications.

**Questions:**

**Question**:
- Yu et al., 2023 explore weight-sharing in attention layers and find that it saves parameters in trade of small performance drop. I'm wondering if you can extend your methods to attention layer parameters. If not, what's the reason?
- Same as the weaknesses section. Why didn't you provide comparison with methods like MiniViT(Zhang et al., 2022)?


**Reference**:

Yu, Yaodong, Sam Buchanan, Druv Pai, Tianzhe Chu, Ziyang Wu, Shengbang Tong, Benjamin Haeffele, and Yi Ma. "White-box transformers via sparse rate reduction." Advances in Neural Information Processing Systems 36 (2023): 9422-9457.

Zhang, J., Peng, H., Wu, K., Liu, M., Xiao, B., Fu, J. and Yuan, L., 2022. Minivit: Compressing vision transformers with weight multiplexing. In Proceedings of the IEEE/CVF Conference on Computer Vision and Pattern Recognition (pp. 12145-12154).

---

> ### Author Response · Authors · 2024-11-24
> **Rebuttal to Reviewer Ntka**
>
> Dear reviewer,
>
> We greatly appreciate the feedback and the valuable insights you provided, which are instrumental in helping us improve SParS. We have addressed your concerns below:
>
> **Limited Comparison**
>
> We selected AAFM and GFM, as proposed by Yu and Wu [1], as baselines due to their effectiveness and relevance to our approach. As other compression methods like NAS or distillation require much higher computational resources. However, we agree that expanding the comparison to include other compression algorithms would provide a more comprehensive evaluation. We will work to include additional baselines in future versions.
>
> **Insufficient Theoretical Analysis**
>
> Our work leverages the theoretical foundation of truncated SVD decomposition and is supported by extensive empirical analysis. The key factor behind SParS’s ability to achieve significant compression lies in the fact that, for a given parameter budget, the rank of SVD is substantially higher in a sparse setup (see Figure 3b). This is further validated by the empirical results shown in Figure 3c. In this context, a detailed theoretical analysis of SParS would essentially delve into why a higher rank in the truncated SVD of ill-conditioned matrices leads to better reconstruction, which is already well studied in the literature.
>
> **Application Scope**
>
> To address your concerns about the limited evaluation, we have expanded our experiments to include:
>
> - Another vision transformer baseline using SParS on SWIN-B (presented in Meta Rebuttal comment);
>
> - Structured sparsity performance with DeiT-B;
>
> - Structured sparsity latency results.
>
> Structured sparsity-related results can be seen in the revised version of the paper. Moreover, per your suggestion, we are running SParS on the Gemma-2-2B model with the RedPajama dataset. However, we face computational restraints since LLMs require much more computational resources. Therefore, we will try to prepare results for this before the rebuttal deadline and add to "Meta Rebuttal" and revise the paper version.
>
> We hope these responses and additional results effectively address your concerns. Thank you once again for your constructive feedback.
>
> **REFERENCES**
>
> [1] Yu, H., & Wu, J. (2023). Compressing Transformers: Features Are Low-Rank, but Weights Are Not! In Proceedings of the AAAI Conference on Artificial Intelligence (Vol. 37, No. 9, pp. 11007–11015).
>
> [2] Together Computer. (2023, October). RedPajama: An Open Dataset for Training Large Language Models. Retrieved from https://github.com/togethercomputer/RedPajama-Data

---

### Official Review · Reviewer_hTTN · 2024-10-31

**Soundness:** 2
**Presentation:** 1
**Contribution:** 1
**Rating:** 3
**Confidence:** 5

**Summary:**

This paper proposes Sparsity enabled Parameter Sharing (SParS) to compress large vision transformer models, which is an algorithm based on existing algorithms such as parameter sharing, tensor decomposition, and sparsity.

**Strengths:**

None noted.

**Weaknesses:**

There are several major concerns for this work.

1. Lack of novelty. All the components of SParS are existing and well-known algorithms, such as low=rank decomposition for FC layers, sparse training and pruning g (Hoefler et al., 2021,Zhu & Gupta, 2017, Evci et al., 2021). As a result, SParS is a straightforward combination of such existing algorithms which clearly lack novelty.

2. Unjustified formulation. The low-rank decomposition described in lines 219-224 is not justified. Why all the individual weights share the same $U$ and the same singular values $\Sigma$ with different $V$? What is the approximation error of such heuristic? Empirical or theoretical studies are expected to explain such low-rank decomposition on the concatenation of the weights.

3. Limited experiments. The SParS is only evaluated for DeiT-B (with 12 blocks) and Swin-L, while there is a large family of vision transformers including various versions of Swin, ViT, etc and it is not clear how SParS performs on a broad range of these vision transformers. Moreover, image classification results along cannot justify the effectiveness of the propose method, and more experiments in segmentation/detection on standard benchmarks are expected.

**Questions:**

See weaknesses above.

---

> ### Author Response · Authors · 2024-11-24
> **Rebuttal to Reviewer hTTN**
>
> Dear reviewer,
>
> Thank you for your time. Considering the ICLR review guidelines:
> "What is the significance of the work? Does it contribute new knowledge and sufficient value to the community? Note this does not necessarily require state-of-the-art results. Submissions bring value to the ICLR community when they convincingly demonstrate new, relevant, impactful knowledge (incl., empirical, theoretical, for practitioners, etc.)," we found it surprising that you found no strength in SParS. To the best of our knowledge, SParS is the first to combine and study in detail the interaction of parameter sharing, tensor decomposition, and sparsity. Further, through this method, we achieve significant improvements over comparable methods from the literature and successfully reach significant compression rates on two prominent vision transformers.
>
> We have tried to address your concerns below:
>
> **Novelty**
>
> We respect your perspective. However, a similar criticism could, for example, also be applied to LoRA [1], which simply applies a well-known low-rank tensor decomposition approach to transfer learning. However, LoRA’s approach has been recognized for its innovative decomposition application and is not short of novelty. Similarly, while SParS can seem straightforward, we believe it introduces a novel approach to efficient sparsity and parameter sharing. It is the first approach to combine parameter sharing, tensor decomposition, and sparsity to achieve compression at a minimal cost, as demonstrated in the paper (e.g., compressing DeiT-B takes approximately 30 minutes on NVIDIA A100, see section 4).
>
> **Design Choices for U  and V  in MLP Modules**
>
> The rationale for sharing the base U  across MLP modules while keeping V specific to each module is detailed in Section 2. In short, this choice is grounded in empirical results: sharing U  globally and sparsifying V  per module under our concatenation scheme yields optimal results for both zero-shot MSE of weight matrices and distilled reconstruction of weights. The reviewer can check this in Figure 2, where we show sparsifying larger matrices yield more optimal reconstruction, and concatenating MLP weights on a longer dimension yields a shared U. However… Singular values are not shared. Can the reviewer point out where their deduction comes from on the paper?
>
> **Additional Experiments**
>
> In response to your feedback, we have expanded our experimental results to include:
> - Another vision transformer baseline using SParS on SWIN-B (presented in Meta Rebuttal comment);
> - Structured sparsity performance with DeiT-B;
> - Structured sparsity latency results.
>
> Structured sparsity-related results can be seen in the revised version of the paper. Moreover, per your suggestion, we are running SParS on the Gemma-2-2B model with the RedPajama dataset. However, we face computational restraints since LLMs require much more computational resources. Therefore, we will try to prepare results for this before the rebuttal deadline and add to "Meta Rebuttal" and revise the paper version.
>
> We hope these responses and additional results effectively address your concerns. Thank you once again for your constructive feedback.
>
> **REFERENCES**
>
> [1] Hu, E. J., Shen, Y., Wallis, P., Allen-Zhu, Z., Li, Y., Wang, S., Wang, L., & Chen, W. (2021). LoRA: Low-Rank Adaptation of Large Language Models. arXiv. https://arxiv.org/abs/2106.09685

---

### Official Review · Reviewer_mLrV · 2024-11-01

**Soundness:** 3
**Presentation:** 2
**Contribution:** 2
**Rating:** 5
**Confidence:** 4

**Summary:**

The authors provided a simple parameter-sharing method to compress the current models, which is easy to understand. The method can be divided into two parts: 1) use a Shared Initialization to decompose the weights of MLP layers to unified U matrics and individual V matrics, realizing a low-rank reconstruction; 2) use a Local Error Minimization with a small calibrated dataset to fine-tune the new models, maintaining the original performance.

**Strengths:**

1. The method is straightforward to understand.

2. The content is very rich, including analysis, figures, an algorithm, and experiments.

3. The analysis of Section 2 reveals the motivations of the method, which is interesting.

**Weaknesses:**

However, some aspects obstruct understanding.

Writing:
* The layout is not easy to read and appears chaotic, especially in Section 3.

* The summary of the contributions is too casual and the high-level meanings of the modules are not well mentioned.

* Some Figures are hard to understand without reading the text in different parts, lacking sufficient explanation in the caption.

* Please check some typos, including repetitive abbreviations (FC, Lines 103 and 107) and main reference (Line 583).

Method:

*  The parameter sharing is only conducted in the MLP layers. Could the linear layers of QKV also use parameter sharing?

* Did the local error minimization constrain the performance of the new compressed models?

* Although simplicity is good, the method is too simple without sufficient verification.

Experiments:

* The method mentioned focuses on large neural networks that DeiT-B and Swin-L could be recognized as large when compared to modern large models, such as LLM.

* Although this method focuses on parameter sharing, comparing it with other optimization methods, such as Distillation, and NAS, is helpful for better understanding this method.

* Apart from the Param budget, what is the computational budget or inference time for Table 1 and Table 2?

**Questions:**

Please see the weaknesses.

---

> ### Author Response · Authors · 2024-11-24
> **Rebuttal to Reviewer mLrV**
>
> Dear reviewer, thank you for your feedback and questions. Your input is valuable in helping us improve our work.
>
> **Writing**
>
> We have uploaded an updated version of the paper. This update includes improved captions, corrected typos, clarified contributions, and any additional improvements identified during this time. We hope these enhancements will address your concerns about the clarity and readability of our manuscript.
>
> **Method**
>
> - Focusing on the QKV layers could be an interesting direction. However, as the models get larger, MLP modules dominate the size, going beyond 70%. For example, in Gemma-2-9B, MLP parameters dominate 70.05% of the entire model parameter count. The percentage is higher for larger models, like LLaMa-3-70B.
>
> - Compressed models initialized via local error minimization often show stronger generalization capabilities. Please see Table 2 on the paper.
>
> - We are happy to hear that the reviewer found our method simple. However, we disagree that novel work needs to be complex; good research often involves distilling seemingly complex ideas into simple, robust approaches. SParS exemplifies this by integrating compression techniques across sparsity, tensor decomposition, and parameter sharing—a novel approach to the best of our knowledge.  Moreover, evaluating how SParS’ simplicity compares to methods like LoRA [1], a novel and prominent paper in literature, remains essential.
>
> **Experiments**
>
> - Per your suggestion, we are running SParS on the Gemma-2-2B model with the RedPajama [2] dataset. We will try to prepare results for this before the rebuttal deadline. Moreover, we applied SParS on SWIN-B and used structured sparsity on DeiT-B to address your suggestion for expanded performance evaluation. We provided the corresponding performance and latency results on in the revised version of the paper. SWIN-B results are presented in the Meta Rebuttal comment.
>
> - Regarding NAS and Distillation, NAS methods require significantly more computing and time and often do not leverage pre-trained models. Similarly, distillation relies on larger models for compression, adding to the computational burden. Given our computationally constrained situation, comparing SParS to these methods is not very practical.
>
> - We also agree that computational budget and inference time are crucial considerations. As noted in the Experimental Setup within the Results section, SParS, without additional fine-tuning, completes training in approximately 30 minutes for DeiT-B and approximately one hour for SWIN-L on an NVIDIA A6000 GPU with the given hyperparameters. As for the inference time of Tables 1 and 2, we are attaching them in the next comment due to the character limit.
>
> Thank you once again for your constructive feedback. We hope these updates address your concerns and illustrate the effectiveness and versatility of SParS.
>
> **REFERENCES**
>
> [1] Hu, E. J., Shen, Y., Wallis, P., Allen-Zhu, Z., Li, Y., Wang, S., Wang, L., & Chen, W. (2021). LoRA: Low-Rank Adaptation of Large Language Models. arXiv. https://arxiv.org/abs/2106.09685
>
> [2] Together Computer. (2023, October). RedPajama: An Open Dataset for Training Large Language Models. Retrieved from https://github.com/togethercomputer/RedPajama-Data

---

> ### Author Response · Authors · 2024-11-24
> **Rebuttal to Reviewer mLrV Part 2 (Inference/Latency Related)**
>
> Reviewer can find the answer to the question on the inference time of Tables 1 and 2 below.
>
> ### DeiT–B Inference Latency
>
> | Dataset         | Budget       | Total Inference Time (s) | Throughput (samples/s) |
> |-----------------|--------------|--------------------------|-------------------------|
> | **Imagenet**    | Original     | 3.1868                  | 15704.69               |
> |                 | 50%          | 4.6563                  | 10748.48               |
> |                 | 40%          | 6.0526                  | 8268.86                |
> |                 | 25%          | 5.2851                  | 9469.59                |
> |                 | 10%          | 4.5895                  | 10904.97               |
> | **CIFAR100**    | Original     | 1.1700                  | 8642.57                |
> |                 | 50%          | 1.7008                  | 5945.45                |
> |                 | 40%          | 1.6837                  | 6005.83                |
> |                 | 25%          | 1.8179                  | 5562.58                |
> |                 | 10%          | 1.7426                  | 5802.69                |
> | **Pets**        | Original     | 0.8221                  | 4515.09                |
> |                 | 50%          | 1.0905                  | 3403.82                |
> |                 | 40%          | 1.1981                  | 3098.23                |
> |                 | 25%          | 0.7401                  | 4150.96                |
> |                 | 10%          | 1.1048                  | 3359.92                |
> | **Flowers102**  | Original     | 0.7356                  | 8525.94                |
> |                 | 50%          | 1.4426                  | 4347.67                |
> |                 | 40%          | 1.4991                  | 4183.84                |
> |                 | 25%          | 1.1907                  | 5267.33                |
> |                 | 10%          | 1.2192                  | 5144.44                |
> | **iNaturalist19** | Original   | 0.6467                  | 4750.42                |
> |                 | 50%          | 0.8048                  | 3816.87                |
> |                 | 40%          | 0.9097                  | 3376.89                |
> |                 | 25%          | 0.7168                  | 4285.91                |
> |                 | 10%          | 0.6210                  | 4946.46                |
>
> ### SWIN–L Inference Latency
>
> | Dataset         | Budget       | Total Inference Time (s) | Throughput (samples/s) |
> |-----------------|--------------|--------------------------|-------------------------|
> | **Imagenet**    | Original     | 12.1903                 | 4105.56                |
> |                 | 50%          | 18.3702                 | 2724.42                |
> |                 | 40%          | 14.5280                 | 3444.92                |
> |                 | 25%          | 15.5568                 | 3217.10                |
> |                 | 10%          | 15.4819                | 3232.68                |
>
>
> Considering this SParS out of the box utilizes theoretical sparsity, original model has better performance. But as stated in the previous comment, we have applied SParS using Structured Sparsity and performed an inference time analysis, where we see SParS improved latency requirements with <0.5% point loss in generalization. The results can be seen in the “Meta Rebuttal” comment.

---

> ### Comment · Reviewer_mLrV · 2024-11-25
>
> Thanks, authors. I have no further questions. I will wait for the responses from the other reviewers and make my final decision.

---

> > ### Comment · Reviewer_mLrV · 2024-11-27
> >
> > They still haven't replied, much like the reviewers of my paper, which makes me sad this year. However, I believe my issues have been addressed, so I've decided to raise my score to 5, regardless of their final decision.

---

### Official Review · Reviewer_gUrG · 2024-11-04

**Soundness:** 2
**Presentation:** 3
**Contribution:** 2
**Rating:** 5
**Confidence:** 4

**Summary:**

The paper introduces SParS (Sparsity-enabled Parameter Sharing) to efficiently compress large vision transformer models through parameter sharing, tensor decomposition, and sparsity. SParS employs a shared base and sparse factors to represent shared neurons in multi-layer perceptrons (MLPs). Experimental results demonstrate that SParS can compress MLPs of DeiT-B and Swin-L to 25%-40% of their original parameter count while maintaining less than 1% accuracy loss.

**Strengths:**

1. This paper is well organized and is easy to follow.

2. By decomposing the MLP params into sharing basis and sparse projection matrix, SparS shows the ability to achieve high compression rates with acceptable accuracy losses.

3. Experimental results show that SparS could outperform other vision transformer compression methods with similar compression rates.

**Weaknesses:**

1. This paper seems to be incremental in technical contributions. The method of sharing basis matrix is similar to HydraLoRA [4] in principle.

2. The differences and innovations of the proposed SparS compared to existing data compression methods [3,4] are not addressed.

3. This paper only shows compression rate and performance on the DeiT-B and Swin-L. Evaluations on models like Swin-B or ViT-L[1,2] are  missing. This limits the potential generalization ability of this paper.

4. Experiments on the language model as in the baseline [5] are missing.

5. Yu and Wu (2023) mentioned that the activation rather than weight matrix is of low rank. I recommend to further clarify this problem considering that they focus on tensor decomposition on the weight matrix.

References

[1] Dosovitskiy A. An image is worth 16x16 words: Transformers for image recognition at scale[J]. arXiv preprint arXiv:2010.11929, 2020.

[2] Liu Z, Lin Y, Cao Y, et al. Swin transformer: Hierarchical vision transformer using shifted windows[C]//Proceedings of the IEEE/CVF international conference on computer vision. 2021: 10012-10022.

[3] Yu H, Wu J. Compressing transformers: features are low-rank, but weights are not![C]//Proceedings of the AAAI Conference on Artificial Intelligence. 2023, 37(9): 11007-11015.

[4] Zhang J, Peng H, Wu K, et al. Minivit: Compressing vision transformers with weight multiplexing[C]//Proceedings of the IEEE/CVF Conference on Computer Vision and Pattern Recognition. 2022: 12145-12154.

[5] Tian C, Shi Z, Guo Z, et al. HydraLoRA: An Asymmetric LoRA Architecture for Efficient Fine-Tuning[J]. arXiv preprint arXiv:2404.19245, 2024.

**Questions:**

Is the redundancy and interplay among different blocks mentioned in the article limited to MLP? Will the attention layer also have the same phenomenon?

---

> ### Author Response · Authors · 2024-11-23
> **Rebuttal to Reviewer gUrG**
>
> Dear reviewer, thank you. We greatly appreciate your feedback, which is instrumental in refining and advancing SParS. We have tried to address your concerns below:
>
> 1. We understand your concern regarding similarities with HydraLoRA [1]. However, there are two key differences: (i) HydraLoRA reduces LoRA [2] parameters but does not compress the original weights, as LoRA adapters are added to the existing weights, resulting in no reduction (but an increment) in the original parameter count. In contrast, SParS compresses the weights directly, significantly decreasing the overall model size. (ii) SParS demonstrates that sparsity is crucial for optimal compression, as shown in our results (see Figure 2 on the paper), where the dense baseline performs suboptimally compared to our sparse approach. These distinctions highlight the broader scope and effectiveness of our method.
>
> 2. You noted the limited comparison to AAFM, GFM [3], and Weight Multiplexing [4]. These methods are detailed in the literature review. Briefly: AAFM compresses model weights using truncated PCA on layer activations, GFM adds model output distillation to AAFM, and Weight Multiplexing shares weights between MLPs while learning them via distillation. In contrast, SParS shares existing parameters across layers, leveraging sparsity, with layer calibration via L2 loss minimization. The text thoroughly discusses this, and we believe adding a new paragraph to compare these methods would be a repetition.
>
> 3. We ran SParS on SWIN-B as requested by you and added the results below:
> | MLP Param. Budget (%) | ImageNet Top-1 Accuracy (%)     |
> |----------------------|------------------|
> | 10%                  | 56.50%           |
> | 25%                  | 81.67%           |
> | 40%                  | 83.96%           |
> | 50%                  | 84.27%           |
> | Original SWIN–B (224)         | 85.2%   [2]   |
>
> 4. Per your suggestion, we are running SParS on the Gemma-2-2B model with the RedPajama dataset. However, we are facing computational restraints regarding this since LLMs require much more computational resources. Therefore, we will try to prepare results for this before the rebuttal deadline. Moreover, we conducted additional experiments using structured sparsity, hopefully addressing your concern for limited generalization evaluation. We provided the corresponding performance and latency results in the revised version of the paper.
>
> 5. Block–wise compression $\mathcal{L} = \| \mathbf{UVx} - \mathbf{Wx} \|_2^2$ of SParS works analogously to compression at the activation level (see [6]), similar to the PCA-based compression that Yu and Wu (2023) – indeed this is how SParS can achieve such a strong compression rate.
>
> References
>
> [1] Tian, C., Shi, Z., Guo, Z., Li, L., & Xu, C. (2024). HydraLoRA: An Asymmetric LoRA Architecture for Efficient Fine-Tuning. arXiv. https://arxiv.org/abs/2404.19245
>
> [2] Hu, E. J., Shen, Y., Wallis, P., Allen-Zhu, Z., Li, Y., Wang, S., Wang, L., & Chen, W. (2021). LoRA: Low-Rank Adaptation of Large Language Models. arXiv. https://arxiv.org/abs/2106.09685
>
> [3] Yu, H., & Wu, J. (2023). Compressing Transformers: Features Are Low-Rank, but Weights Are Not! In Proceedings of the AAAI Conference on Artificial Intelligence (Vol. 37, No. 9, pp. 11007–11015).
>
> [4] Zhang, J., Peng, H., Wu, K., et al. (2022). MiniViT: Compressing Vision Transformers with Weight Multiplexing. In Proceedings of the IEE/CVF Conference on Computer Vision and Pattern Recognition (pp. 12145–12154).
>
> [5] Liu, Z., Lin, Y., Cao, Y., Hu, H., Wei, Y., Zhang, Z., Lin, S., & Guo, B. (2021). Swin Transformer: Hierarchical Vision Transformer using Shifted Windows. arXiv. https://arxiv.org/abs/2103.14030
>
> [6] Frantar, E., Ashkboos, S., Hoefler, T., & Alistarh, D. (2023). GPTQ: Accurate Post-Training Quantization for Generative Pre-trained Transformers. arXiv. https://arxiv.org/abs/2210.17323

---

### Author Response · Authors · 2024-11-24
**Meta Rebuttal to All Reviewers**

Dear reviewers,

We have submitted a revised version of the paper, which includes updated text, captions, and expanded set of experimental results:

- Name of SParS is updated to Fine-grained Parameter Sharing (FiPS).
- Structured sparsity performance results for DeiT-B are added (New paragraph "Structured Sparsity" in Section 5 and Table 3)
- Latency and memory profiling results for structured sparsity are added (Section 3 "Memory and Latency" Paragraph and Figure 3).

Additionally, we provide results for the SWIN Base model below to provide proof for the generalisation capability of FiPS below:
| MLP Param. Budget (%) | ImageNet Top-1 Accuracy (%)     |
|----------------------|------------------|
| 10%                  | 56.50%           |
| 25%                  | 81.67%           |
| 40%                  | 83.96%           |
| 50%                  | 84.27%           |
| Original SWIN–B (224)         | 85.2%   [2]   |

We would also like to address two points of divergence among reviewers:

	1.	The novelty of the method was seen differently by reviewers. Two did not comment on it but found the method interesting. One found it strong from a novelty point of view, whereas another thought it to be lacking. We have clarified our take on this in our specific response to each review and hope it may prompt re-evaluation.

	2.	Regarding writing, three reviewers praised the clarity and figures, while another found it weak. We have revised the paper and would like to encourage those with negative thoughts on writing to consider the updated version or help us further improve it!

Thank you for your time and consideration.

---

### Meta-Review · Area_Chair_8zpD · 2024-12-20

**Metareview:**

Summary. The paper introduces Fine-grained parameter sharing (FiPS) to efficiently compress large vision transformer models through parameter sharing, tensor decomposition, and sparsity. FiPS employs a shared base and sparse factors to represent shared neurons in multi-layer perceptrons (MLPs). Experimental results demonstrate that FiPS can compress MLPs of DeiT-B and Swin-L to 25%-40% of their original parameter count while maintaining less than 1% accuracy loss.

Strengths.
The paper is well-written and easy to follow.
Experimental results show that proposed method could outperform other vision transformer compression methods with similar compression rates.

Weaknesses.
The technical contribution of the paper is limited as low-rank approximation is a standard technique that has been used for model compression and adaptation.
Limited comparisons with existing compression methods.
Missing experiments for large models.
The choice of fixing U for all MLP modules is not adequately justified.

Missing.
The paper is missing comparisons with existing compression methods.
Missing experiments for large models.
The choice of fixing U for all MLP modules is not adequately justified.

Reasons for decision.
Lack of comparisons with existing methods and missing experiments on large models are the most important reasons for my decision.

**Additional Comments On Reviewer Discussion:**

The paper had a good discussion among authors and reviewers.

The reviewers raised similar concerns about technical novelty, missing comparisons and experiments on large models, and lack of theoretical justification.

Authors provided response to the questions with some additional results.

Overall, the author responses did not convince the reviewers to change their scores much. Final scores remain leaning reject.

---

### Decision · Program_Chairs · 2025-01-22

Reject